# Modulation of Gut Microbiome and Autism Symptoms of ASD Children Supplemented with Biological Response Modifier: A Randomized, Double-Blinded, Placebo-Controlled Pilot Study

**DOI:** 10.3390/nu16131988

**Published:** 2024-06-21

**Authors:** Jela Hrnciarova, Klara Kubelkova, Vanda Bostik, Ivan Rychlik, Daniela Karasova, Vladimir Babak, Magdalena Datkova, Katerina Simackova, Ales Macela

**Affiliations:** 1Faculty of Medicine, Charles University, 500 03 Hradec Kralove, Czech Republic; hrnciarovaj@lfhk.cuni.cz (J.H.); datkovam@lfhk.cuni.cz (M.D.); katerina.simackova@fnhk.cz (K.S.); 2Department of Psychiatry, University Hospital in Hradec Kralove, 500 03 Hradec Kralove, Czech Republic; 3Military Faculty of Medicine, University of Defence, 500 03 Hradec Kralove, Czech Republic; vanda.bostikova@unob.cz (V.B.); amacela@seznam.cz (A.M.); 4Veterinary Research Institute, 621 00 Brno, Czech Republic; ivan.rychlik@vri.cz (I.R.); daniela.karasova@vri.cz (D.K.); vladimir.babak@vri.cz (V.B.)

**Keywords:** autism, microbiome, biological response modifier, psychobiotics

## Abstract

The etiology and mechanisms of autism and autism spectrum disorder (ASD) are not yet fully understood. There is currently no treatment for ASD for providing significant improvement in core symptoms. Recent studies suggest, however, that ASD is associated with gut dysbiosis, indicating that modulation of gut microbiota in children with ASD may thus reduce the manifestation of ASD symptoms. The aim of this pilot study (prospective randomized, double-blinded, placebo-controlled) was to evaluate efficacy of the biological response modifier Juvenil in modulating the microbiome of children with ASD and, in particular, whether Juvenil is able to alleviate the symptoms of ASD. In total, 20 children with ASD and 12 neurotypical children were included in our study. Supplementation of ASD children lasted for three months. To confirm Juvenil’s impact on the gut microbiome, stool samples were collected from all children and the microbiome’s composition was analyzed. This pilot study demonstrated that the gut microbiome of ASD children differed significantly from that of healthy controls and was converted by Juvenil supplementation toward a more neurotypical microbiome that positively modulated children’s autism symptoms.

## 1. Introduction

Autism spectrum disorder (ASD) is a behaviorally defined neurodevelopmental disorder. ASD lacks specific clinical biomarkers and has seen an evolving conceptualization through the decades since it was first described. Over the past four decades, there has been dramatic increase in the number of individuals diagnosed with ASD [1]. In general, ASD is diagnosed by 3 years of age in most of those children experiencing it, although roughly 40% of such children are not first evaluated until 4 years of age [2]. A psychiatric diagnosis of ASD, which has behavior as its basis of definition, relies heavily upon precise observation and clinical expertise because the condition lacks standardized biomarkers [3].

ASD is one of the most common and challenging neurodevelopmental disorders in children. Its prevalence rate worldwide now exceeds 1%. A small number of these children appear to develop normally in their first year and then go through a period of regression between 18 and 24 months of age. ASD is characterized by deficits in communication and social interaction, as well as a presence of repetitive and restrictive behaviors. Moreover, ASD often manifests with a wide range of comorbidities that include morphological, physiological, and psychiatric conditions. ASD’s most commonly proposed causes are physiological and metabolic disorders involving immunity, oxidative stress, and mitochondrial dysfunction [4]. Co-occurrence of two or more disorders in the same individual has been observed, with comorbidities including anxiety, depression, attention deficit/hyperactivity disorder (AD/HD), epilepsy, gastrointestinal symptoms/problems, sleep disorders, learning disabilities, obsessive–compulsive disorder, intellectual disability, sensory problems, and immune disorders. The most prevalent comorbidity, at roughly 50%, is intellectual disability [5]. At least one comorbidity exists in about 70% of children with ASD, while 41% have two or more [6]. An estimated 20% of individuals diagnosed with ASD also have epilepsy [7].

Although ASD’s etiology remains largely unexplained, a recent finding identifies specific gut microbiota composition in ASD patients. Post-mortem examination of ASD subjects’ brain tissue and small intestines has revealed that the blood–brain barrier and gut barrier were disrupted, with significant neuroinflammation evidenced by increased expression of genes and markers associated with brain inflammation. It has further been inferred that the gut–brain axis disruption may be associated with non-self antigens that trigger a neuroinflammatory reaction by crossing the damaged gut barriers, thus leading to ASD in genetically susceptible subjects [8]. In a study involving 192 twins, however, genetic factors accounted for only 38% of ASD risk, whereas the remaining 68% was attributed to environmental factors [9]. A significant role was ascribed to the gut microbiota [10]. Microbiota is shaped by diet, lifestyle, and microbial exposure in the early developmental phase and infection, as well as by genetic makeup, metabolites, and immunological and hormonal aspects [11].

Analysis of gut microbiota is currently a growing area of research linked to neuropsychological disorders, including depression [12], metabolic disorders such as obesity [13], and gastrointestinal disorders, including inflammatory bowel disease or irritable bowel syndrome [14,15]. Many studies have identified that microbiota composition in ASD patients differs significantly from that in healthy controls [16,17,18,19]. Nutritional intervention, prebiotics, probiotics, and symbiotics, including fecal microbiota transplantation as a remedy to modulate the species composition of the intestinal microbiota of patients with ASD, already have been tested [20,21,22,23,24]. These studies generally have concluded, however, that their findings should be taken with caution because there still exist only limited data from studies examining different regimens of different remedy applications and there have not yet been double-blind studies demonstrating clinical significance of the effects of those remedies used. Here, therefore, we present the results of a double-blind, placebo-controlled study utilizing the biological response modifier Juvenil for influencing the gut microbiota of children suffering from ASD and for modulating their ASD symptoms.

## 2. Study Design, Materials, and Methods

### 2.1. Study Design

The pilot prospective double-blind randomized feasibility study enrolled 28 children (9 girls and 19 boys) of Czech nationality aged 3 to 7 years. Of these, 16 children under care of the Psychiatric Clinic of the Hradec Kralove University Hospital, Czech Republic met the criteria for a diagnosis of ASD and the remaining 12 children (control group) were neurotypical (NT), i.e., without any signs of ASD (Table 1). The ASD children were randomly selected for the study by their attending psychiatrists. The children forming the control (neurotypical) group were randomly included in the study on the basis of an agreement with parents living in the geographical area of this study. These children were never under the care of a psychiatrist.

The group of autistic children was randomly divided into two groups of 8 children each. The first group was administered Juvenil, while the second group of 8 autistic children was given a placebo throughout the study. Juvenil or placebo capsules were administered orally to the children by their parents at home once a day for 3 months. The inclusion of children with autism into the study did not affect their existing treatments, education, or rehabilitation. To evaluate the effect of Juvenil on the gut microbiome, stool samples were collected once from healthy children and from autistic children before and after providing Juvenil or placebo. Autistic children were evaluated using the Childhood Autism Rating Scale in its standard version (CARS2-ST) by a clinical psychiatrist and subjective information based on observations of the children’s behavior was obtained by conducting interviews with the children’s parents. The study was approved by the hospital’s ethics committee and informed consent was signed by the parents of all study participants (see Figure 1).

### 2.2. Juvenil and Placebo

The Juvenil and placebo capsules were prepared by Uniregen, Ltd., Nachod, Czech Republic. Capsules of Juvenil contained 1.0 mg of Juvenil crude substance while placebo capsules contained redistilled water. Both capsule types looked identical and were stored at room temperature until use.

Juvenil is a nontoxic alcohol–ether extract of bovine tissue registered as a dietary supplement with modulatory activity on the immunity, dominantly on cells of the innate immune system, and on the organism’s regeneration [25,26]. Juvenil is a complex mixture of peptides, nucleotides, free amino acids, and some other components of animal origin [27].

### 2.3. Stool Samples Procedure

The following recommendations were given to parents concerning fecal sample collection: (1) sample a stool from a clean container (e.g., potty) or from a piece of stool on toilet paper; (2) the stool must not be diluted with water or urine; (3) collect the stool sample using the scoop placed in the lid of the collection container; (4) a small amount of stool is sufficient for the examination (i.e., not larger than the size of a hazelnut or 1–2 mL in the case of a liquid stool sample); (5) return the sampling scoop to the container and screw on the cap; (6) place the collection container in a plastic bag, close the bag, and label it with the child’s name and surname, not marking the sample container in any other way; (7) place the container with the stool sample in a closed plastic bag and keep it at 4 °C (in the refrigerator); (8) personally deliver the collection container with the sample to the doctor no later than 24 h after taking the sample (preferably the same day, but no later than the following day); and (9) stool samples should be collected one day before the start of supplementation and on the last day of supplementation (i.e., after 12 weeks from the start of supplementation).

The collection container and its contents were labeled by the attending physician and the samples were stored at −20 °C until subsequent examination.

### 2.4. Microbiota Analysis

Microbiota composition was determined as described previously [28]. The samples were homogenized in a MagNALyzer (Roche, Basel, Switzerland). Following homogenization, the DNA was extracted using a QIAamp DNA Stool Mini Kit (Qiagen, Hilden, Germany) according to the manufacturer’s instructions and the DNA concentration was determined spectrophotometrically. DNA samples were diluted to 5 ng/mL and were used as template in polymerase chain reaction (PCR) with forward primer 5′-TCGTCGGCAGCGTCAGATGTGTATAAGAGACAG-MID-GTCCTACGGGNGGC WGCAG-3′ and reverse primer 5′-GTCTCGTGGGCTCGGAGATGTGTATAAGAGACA G-MIDGTGACTACHVGGGTATCTAATCC-3′. MIDs shown above represent different sequences 5, 6, 7, or 9 base pairs in length that were used to identify individual samples within the sequencing groups. PCR amplification was performed using a HotStarTaq Plus Master Mix kit (Qiagen) and the resulting PCR products were purified using AMPure beads (Beckman Coulter, Prague, Czech Republic). In the next steps, the concentration of PCR products was determined spectrophotometrically, the DNA was diluted to 100 ng/µL, and groups of 14 PCR products with different MID sequences were indexed with the same indices using a Nextera XT Index Kit (Illumina, San Diego, CA, USA). Prior to sequencing, the concentrations of differently indexed samples were determined using a KAPA Library Quantification Complete kit (Kapa Biosystems, Wilmington, MA, USA), all indexed samples were diluted to 4 ng/µL, and 20 pM phiX DNA was added to final concentration of 5% (*v*/*v*). Sequencing was performed using a MiSeq Reagent Kit v3 and MiSeq apparatus (Illumina).

Sequencing data were analyzed using QIIME 2 [29]. Raw sequence data were demultiplexed and quality filtered; sequencing primers were then clipped using Je [30] and Fastp [31]. The resulting sequences were denoised with DADA2 [32]. Taxonomy was assigned to ASVs using the q2-feature-classifier [33] classify-sklearn naïve Bayes taxonomy classifier against the Silva 138 [34]. All the software tools were used with default settings.

### 2.5. Statistical Methods

Microbiota composition in different groups of patients was compared using permutational multivariate analysis of variance (PERMANOVA, R project, package vegan, function adonis2; Bray–Curtis dissimilarity, 9999 permutations). If PERMANOVA rejected the null hypothesis, then pairwise comparisons were made of all groups. Statistical significance was established at *p* < 0.05. LEfSe (linear discriminant analysis effect size) was used to determine taxa which most likely explained the differences between the compared groups [35]. To correlate the individual categories of ASD patient symptoms with bacterial taxa that may characterize the gut dysbiosis of children with ASD, the Covariance S tool in Microsoft Excel 2021 (v16.0) was used.

## 3. Results

### 3.1. Gut Microbiota Composition in Autistic and Neurotypical Children

Comparison of beta diversity using principal coordinate analysis (PCoA) confirmed differences in the clustering of samples from control and ASD children (Figure 2). All groups of ASD patients (NT, ASD Placebo, and ASD before and after Juvenil treatment) harbored microbiota different from those of the NT group (*p* = 0.001). At the bacterial phylum level, there were significant differences in the abundance of Actinobacteriota, Firmicutes, and Proteobacteria. While Actinobacteriota and Proteobacteria dominated in autistic children, Firmicutes were more abundant in neurotypical controls (Table 2 and Appendix A).

LEfSe analysis (linear discriminant analysis effect size) identified bacterial taxa discriminating the ASD groups prior to supplementation from the NT group (Figure 3). *Bifidobacterium longum, Ruminococcus torque, Faecalibacterium, Flavonifractor*, *Pseudomonadas*, and *Clostridia* vadinBB60 were characteristic for the gut microbiota of autistic children while *Streptococcus parasanguinis, Monoglobus*, *Terrisporobacter*, or *Bacteroides cellulosilyticus* were more abundant in the microbiota of healthy controls.

### 3.2. Gut Microbiome Modulation by Juvenil

Pairwise comparison of NT, ASD patients before treatment, and ASD patients after Juvenil supplementation demonstrated there to be a significant difference in the microbiota composition between the NT and ASD patients before supplementation, but there was no significant difference in microbiota composition in a comparison of ASD patients after Juvenil either with NT healthy controls or with ASD patients before treatment. Juvenil administration thus shifted the profile of the gut microbiota composition of autistic children toward that of the neurotypical children, although it did not result in the restoration of a completely healthy type of microbiota. In addition, there were no significant differences in microbiota composition at the phylum level when comparing ASD patients either before and after placebo treatment or before and after Juvenil administration (Table 3).

A moderate effect of Juvenil administration in comparison to placebo can be seen also from the comparison of operational taxonomic units (OTUs) completely lost or newly acquired during treatment (Appendix A). The greatest losses or acquisition were recorded in OTUs belonging to families *Lachnospiraceae*, *Ruminococcaceae*, *Oscilospiraceae*, and *Christenellaceae*, all comprising spore-forming bacterial species. While children after placebo treatment lost 305 OTUs and newly acquired 132 OTUs, microbiota in Juvenil-treated ASD patients were more stable as there were only 200 OTUs lost and 152 newly appearing (Appendix A).

### 3.3. Behavioral Status of Autistic Children and Juvenil

The behavioral status of autistic children of both groups was assessed using CARS2-ST at the time of entry into the study and after completion of supplementation. A comparison of the group supplemented with Juvenil and the group supplemented with placebo showed a positive shift in the values of the rating scale parameters during 3-month supplementation with Juvenil (Table 4 and Appendix A). Nevertheless, the changes associated with Juvenil as well as placebo supplementation did not reach statistical significance (ASD placebo group *p* = 0.62, ASD Juvenil group *p* = 0.19). There were also no significant differences in the effect of supplementation in children with mild symptoms of ASD and severe symptoms of ASD in either group (Juvenil group *p* = 0.95, placebo group *p* = 0.82). Numerically, however, using the percentile parameter, there was a 12.4% reduction in autism symptoms associated with Juvenil supplementation, which was approximately double that associated with the placebo (6.6% reduction). In comparing the difference between Juvenil supplementation versus placebo using the T-score parameter, we can observe shifts of two or more points in favor of Juvenil for categories 4 (motor manifestations), 7 (visual reactions), 10 (fear and nervousness), 12 (nonverbal communication), and 13 (activity level). In addition, Juvenil showed a significant positive effect (*p* = 0.009) when comparing the index of individual CARS-2 ST categories modulation by Juvenil and placebo, respectively (Table 4).

### 3.4. Correlation between Abundance of Key Bacterial Genera and ASD Symptoms

To further explore whether specific microbiota composition can be associated with ASD symptoms, the abundance of 22 genera reported by other authors as associated with ASD symptoms were compared with results from CARS2-ST testing using a covariance S test (Figure 4). *Prevotella*, *Escherichia/Shigella*, *Veillonella*, *Streptococcus*, *Alistipes*, and *Bifidobacterium* had the highest positive correlation coefficients in relation to the total CARS2-ST score (i.e., the greater the abundance of these genera, the more severe were the autism symptoms). On the other hand, a negative correlation was observed for *Bacteroides*, *Faecalibacterium*, *Barnesiella*, and *Blautia* (Figure 4A), indicating that an increase in the abundance of these genera was associated with relief in autistic symptoms. Of the five tested CARS2-ST categories, the nonverbal communication category was characterized by a positive correlation with *Blautia* (Figure 4E). For all categories tested, including the total score, *Veillonella*, *Streptococcus*, and *Clostridium* repeatedly exhibited a positive correlation.

## 4. Discussion

The gut microbiota is a complex ecosystem that, through its metabolites or enteroendocrine cell products induced by those metabolites, affects the microbiota–gut–brain axis and thus homeostasis of the entire organism. Consistent with previously published studies [36,37,38,39,40,41], a different composition of gut microbiota was recorded for children with ASD compared to unrelated neurotypical controls. The significant changes in representation were demonstrated in the phyla *Actinobacteriota*, *Firmicutes*, and *Proteobacteria*. LEfSe analysis revealed an increased abundance of *Bifidobacterium longum*, *Ruminococcus torques* group, *Faecalibacterium*, *Flavonifractor*, and several taxa of *Clostiridiae* and *Pseudomonodaceae* as characteristic for the gut microbiota of autistic children. On the other hand, autistic children were characterized also by lower abundance of taxa from the *Prevotellaceae*, *Peptostreptococcaceae*, and *Monoglobaceae* families. *Faecalibacterium prausnitzii* (*F. prausnitzii*) is one of the main producers of butyrate in the intestine and, because butyrate is an inhibitor of NF-κB and IFN-γ [42], *F. prausnitzii* may interfere with the body’s inflammatory responses. Moreover, *F. prausnitzii* is an of IL-10 inducer and may, therefore, be referred to as an anti-inflammatory gut bacterium [43]. Aside from *Faecalibacterium*, the remaining positively scored bacterial taxa (e.g., *Ruminococcus*, *Flavonifractor*, and *Bifidobacterium* species have been suggested as predictors of more adverse post-traumatic neuropsychiatric sequelae outcomes [44]. *Flavonifractor* prevalence has also been associated with another psychiatric diagnosis of affective disorder [45]. On the other hand, the bacterial taxa with decreased abundance in the gut microbiota of autistic children (e.g., *Prevotella*) have high genetic diversity and, therefore, it is difficult to predict their functional relationships to autism [46,47]. Whether *Prevotella* is or is not beneficial to health depends on many factors [48], so it cannot be used unambiguously as a predictive factor of gut dysbiosis in autism [49]. A lower abundance of the families *Monoglobaceae* and *Peptostreptococcaceae* in the gut microbiome has been associated with maternal prenatal stress or anxiety symptoms [50,51].

In this pilot study, the genuses *Bacteroides* and *Prevotella* were found to have the highest negative and positive correlation coefficients, respectively, in relation to total CAR2-ST score. *Bacteroides* and *Prevotella*, two quite closely related genera, were frequently associated with extreme opposite autism symptoms (Figure 4). Interestingly, *Prevotella* is usually enriched in African ethnics with a high proportion of plants in their food (enterotype 2) while *Bacteroides* enrichment is associated with a Western diet (enterotype 1) [52,53]. Another interesting observation for *Prevotella* and *Bacteroides* is that *Bacteroides* dominates the gut microbiota of piglets or humans under lactation and is replaced within a short time after weaning by the related *Prevotella* [54,55,56]. This may point to the importance of weaning, diet, and associated changes in the gut microbiota for the development of autism in children. In this respect, moreover, the mother’s diet during pregnancy and lactation, especially from the viewpoint of consuming a Western diet with an unbalanced ratio of polyunsaturated fatty acids, may adversely affect brain development [57,58].

Microbiota transfer therapy or probiotic supplementation of ASD individuals has been tested as means to alleviate ASD symptoms by modulation of gut microbiota [59,60,61,62,63,64,65,66]. Microbiota transfer therapy treatment protocol, consisting of the application of vancomycin, the laxative MoviPrep, SHGM (Standardized Human Gut Microbiota), and Prilosec (Omeprazole), reduced the rates of core ASD symptoms [5,64]. The administration of probiotics, either alone or in combination with other biologically active substances such as colostrum or oxytocin, led to minor reductions in autism symptoms but without reaching statistical significance [62,63,67]. Only a study that was based on the application of four probiotic bacteria in combination with fructooligosaccharides provided a significant reduction in the severity of autism and gastrointestinal symptoms [65]. Although our results are in agreement with these studies, in that a biologically active substance may alleviate ASD symptoms, whether this is a direct immunostimulating effect or rather is caused by modification of the gut microbiota composition, remains uncertain.

Data from this pilot study presented here demonstrate significant changes in the microbiome, and in parallel, the effects on some categories of CARS2-ST. However, these results must be approached with caution, as children in home care were included in this study, and confounding variables such as diet in the family, passive smoking, the influence of complementary medicines used by children, or the physical and psychological family environment could have impacted microbiota profiles.

## 5. Conclusions

This pilot study confirms, albeit in a small number of children, that children with ASD have altered composition of the gut microbiota. A high abundance of *Bacteroides* was associated with weaker ASD symptoms while *Prevotella*, *Escherichia/Shigella*, *Veillonella*, *Streptococcus*, *Alistipes*, and *Bifidobacterium* were enriched in the gut microbiota of autistic children with strongest symptoms. An altered composition of ASD children’s gut microbiota was shifted toward a neurotypical profile by Juvenil supplementation. Juvenil also positively modulated children’s autism symptoms, namely in the categories of motor manifestations, visual reactions, fear and nervousness, nonverbal communication, and activity level. Juvenil supplementation of ASD children was safe, well-tolerated, and had no side effects.

## Figures and Tables

**Figure 1 nutrients-16-01988-f001:**
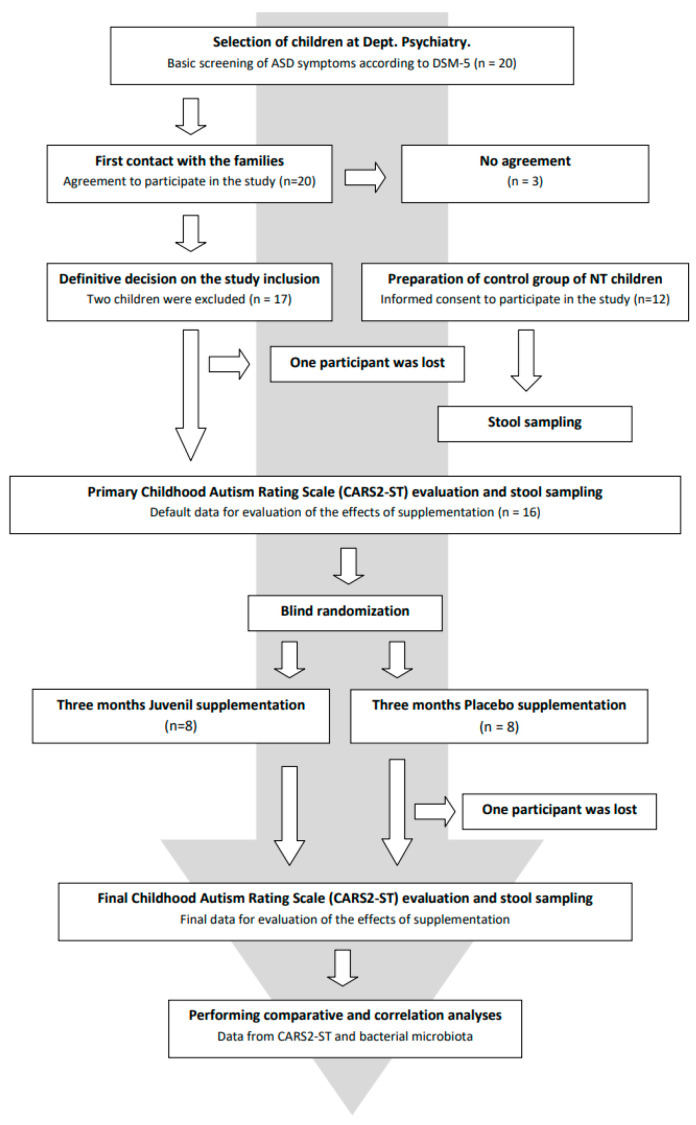
Flow diagram of the participant progress from recruitment to the end of the experiment. Include exclusions/dropouts.

**Figure 2 nutrients-16-01988-f002:**
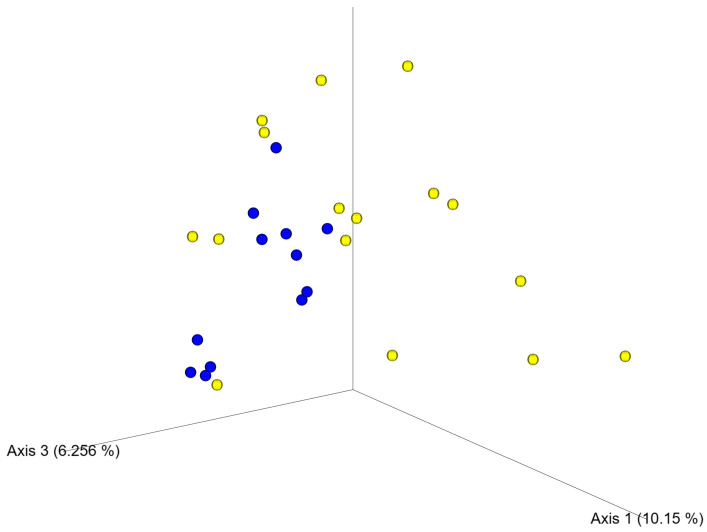
Fecal microbiota analysis of ASD and control children. Principal coordinate analysis (PCoA) using Bray–Curtis distance matrix separated samples of control and ASD children. Blue dots indicate samples from neurotypical control children; yellow dots indicate samples from ASD children.

**Figure 3 nutrients-16-01988-f003:**
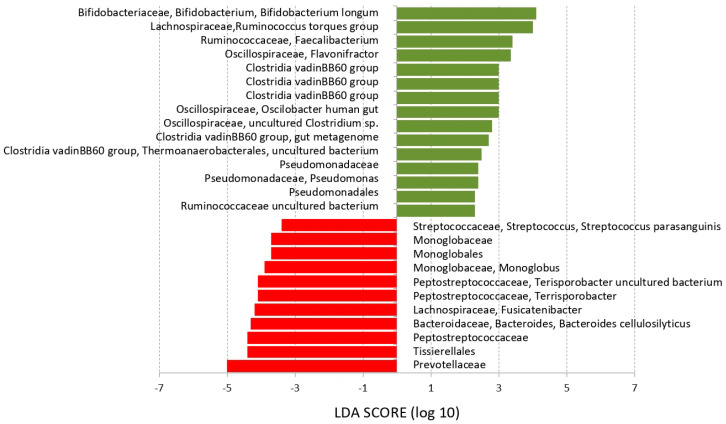
Taxa characteristic for ASD and control children. LEfSe analysis identified bacterial taxa in stool samples typical for ASD (green) and neurotypical control children (red).

**Figure 4 nutrients-16-01988-f004:**
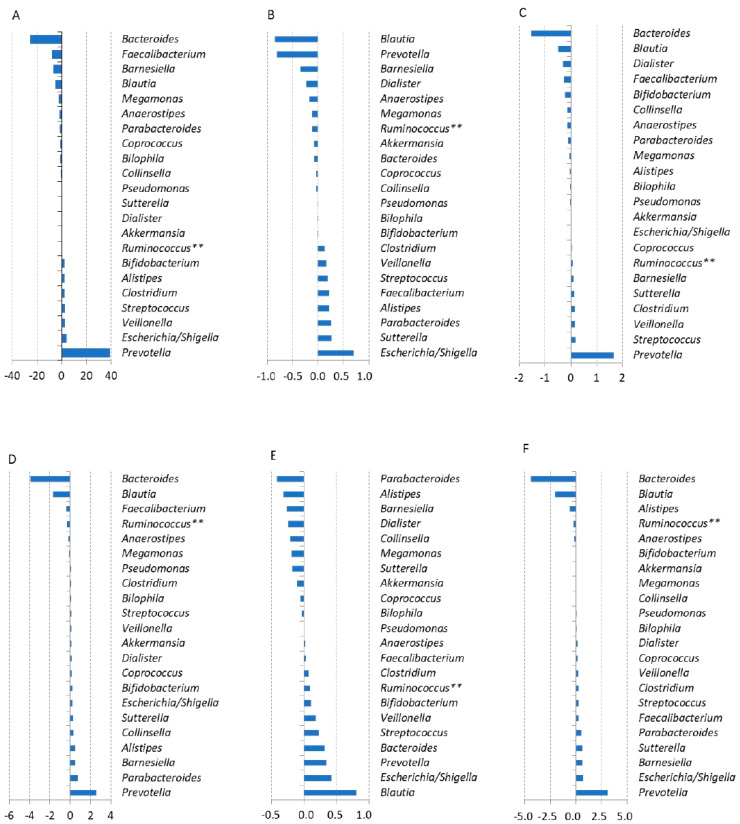
Comparison of microbiome data with total scores of CARS2-ST and its individual categories in which the total score shifted by at least two points after Juvenil supplementation. (**A**)—total score, (**B**)—body, (**C**)—visual response, (**D**)—fear and nervousness, (**E**)—nonverbal communication, (**F**)—activity level. ** only *Ruminococcus torques* group and/or *Ruminococcus gauvreauii*_group.

**Table 1 nutrients-16-01988-t001:** Demographic characteristics of the subjects involved in this study.

Characteristic	Value	
	ASD Group	Neurotypical Group
Average age at enrollment (years)	6 ± 3 *	5 ± 2 *
Age range (years)	3–9	3–9
Male/Female, (number)	13/3	8/4
Ethnicity/Location	White/East Bohemia, CZE	White/East Bohemia, CZE

Note: * mean ± SD.

**Table 2 nutrients-16-01988-t002:** Relative abundance of gut microbiota between ASD group (without any treatment) and neurotypical control group (NT) at phylum levels.

Phylum	ASD	NT	*p*-Value
*Actinobacteriota*	3.04	1.18	0.03
*Bacteroidota*	49.02	49.48	0.60
*Campylobacterota*	0.03	0	0.35
*Cyanobacteria*	0.09	0.12	0.92
*Desulfobacterota*	0.29	0.17	0.39
*Euryarchaeota*	0.006	0.21	0.03
*Firmicutes*	41	46	0.004
*Fusobacteriota*	0.001	0.003	0.17
*Patescibacteria*	0.02	0.002	0.08
*Proteobacteria*	6	2	0.01
*Synergistota*	0.005	0	0.28
*Verrucomicrobiota*	0.84	0.69	0.80

**Table 3 nutrients-16-01988-t003:** Comparison of microbiota composition (%) at the phylum level before and after Juvenil or placebo treatment.

Phylum	Juvenil			Placebo		
	Before	After	*p*-Value	Before	After	*p*-Value
*Actinobacteriota*	3.78	2.25	0.26	2.78	2.54	0.84
*Bacteroidota*	51.05	58.62	0.57	50.83	57.22	0.53
*Campylobacterota*	0.07	0.06	0.91	0.0006	0	0.32
*Cyanobacteria*	0.07	0.06	0.83	0.05	0.11	0.36
*Desulfobacterota*	0.29	0.13	0.48	0.33	0.36	0.79
*Euryarcheota*	0.005	0.02	0.37	0.01	0.02	0.54
*Firmicutes*	36.37	34.05	0.60	41.83	34.29	0.12
*Fusobacteriota*	0.0008	0	0.32	0.0009	0	0.32
*Patescibacteria*	0.006	0.006	0.95	0.02	0.01	0.44
*Proteobacteria*	6.97	4.11	0.22	3.48	4.38	0.59
*Synergistota*	0.005	0.002	0.56	0.007	0	0.32
*Verrucomicrobiota*	1.28	0.61	0.48	0.66	1.07	0.51

**Table 4 nutrients-16-01988-t004:** Comparison of individual CARS2-ST * categories in Juvenil- and placebo-supplemented groups. Data were collected before and after 3-months supplementation. Before and after numbers are sums of the CARS2-ST category values from all members of the given group (Juvenil, placebo).

	Childhood Autism Rating Scale (CARS2-ST)	Σ Juvenil		Σ Placebo		Shift	Index	Index
	Category	Before	After	Before	After	Juvenil/Placebo	Juvenil	Placebo
1	Relationship to people	26.5	24.5	21.5	19.5	2/2	0.916	0.921
2	Imitation	23.0	21.5	17.5	16.0	1.5/1.5	0.928	0.933
3	Emotional response	21.5	19.5	22.5	22.0	2/0.5	0.902	1
4	Body	21.0	19.0	20.0	20.0	2/0	0.900	1
5	Object use	20.5	19.0	15.5	15.0	1/0.5	0.921	1
6	Adaptation to change	19.5	18.5	21.5	19.5	1/2	0.944	0.916
7	Visual response	19.5	17.5	18.5	18.5	2/0	0.892	1
8	Listening response	22.0	21.0	17.5	17.5	1/0	1	1
9	Taste–smell–touch response and use	18.0	18.0	18.0	17.0	0/1	1	1
10	Fear and nervousness	20.0	17.0	20.0	19.5	3/0.5	0.823	0.972
11	Verbal communication	26.5	26.5	24.5	23.0	0/1.5	1	0.952
12	Nonverbal communication	24.0	22.5	16.5	18.0	1.5/−1.5	0.930	1
13	Activity level	24.0	22.0	21.5	21.5	2/0	0.904	1
14	Level and consistency of intellectual response	21.5	21.5	24.5	24.5	0/0	1	1
15	General impressions	25.0	24.5	24.0	24.0	0.5/0	0.978	1
	*t*-test						*p* = 0.0095	

Note: * Eric Schopler, Mary E. Van Bourgondien, Glenna Janette Wellman, Steven R. Love. (CARS™2) Childhood Autism Rating Scale™, Second Edition (https://www.wpspublish.com/cars-2-childhood-autism-rating-scale-second-edition.html, accessed on 12 May 2020).

## Data Availability

All data are available within the manuscript.

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
