# Peer review of "Modulation of Gut Microbiome and Autism Symptoms of ASD Children Supplemented with Biological Response Modifier: A Randomized, Double-Blinded, Placebo-Controlled Pilot Study"

_nutrients, 2024, doi:10.3390/nu16131988_

Round 1

Reviewer 1 Report

Comments and Suggestions for Authors

The study described in the article introduces new information in the field of supplementation and modification of intestinal microbiota beneficial in the course of ASD. The greatest concern, however, is the number of patients tested, especially in the control group. I understand that the center may not have the funds to increase the number of study groups.

1. the study group is small, the study should be treated as a pilot study, which should also be indicated in the title of the article

2. the consent of the ethics committee along with its number and date of issue, as well as information about informed consent, should be included at the end of the article in the appropriate section. Duplicate information should be removed in the material and methods section

3. Table 1 is missing a unit for the average age (years)

4. missing unit in table 3

5. Figure 2 is blurry and needs to be improved

6. It would be good to include information about a possible conflict of interest and how to obtain or purchase Juvenil from the manufacturer. no benefits for co-authors from the research (financial, travel grants, gifts, etc.)

the article is carefully written and in understandable language, minor comments should be corrected

Reviewer 2 Report

Comments and Suggestions for Authors

The effort to identify treatments for ASD is worthwhile and complementary/alternative medicine is an option to traditional pharmaceuticals. The treatment, Juvenil is reported here to offer benefit for ASD symptoms. The manuscript would be improved with further details provided on the treatment and the participants.

Abstract: Please describe the number of participants and duration of treatment.

Methods: Describe recruitment of the participants. How were parents notified of the study? Were control children also under psychiatric care? What were the diagoses of the controls?

Table 1: Move to Results and include Age of enrollment, mean (SD)

Describe the randomization process.

Please create a flow diagram of the participant progress from recruitment to end of experiment. Include exclusions/drop-outs.

Was Child Assent in addition to Parental Consent obtained?

Provide a citation for CARS2-ST.

You appropriately provide references to your earlier work, but it would be helpful to understand more about the composition of Juvenil. A table showing nutrient/bioactive/chemical analysis would be valuable.

Results:

It appears that important variables were not included in analysis. I understand this is a pilot study, but factors such as diet and medications have established impacts on microbiota profile and behavior.  As mentioned above, the psychiatric disorders of the Controls is also important to consider.

Explain in statistical analysis how CARS2-ST (line 238, Table) was calculated and what it means. Check CARS-2 ST vs CARS2-ST.

You report that no adverse effects were seen. What data were collected regarding side-effects?

Discussion: How do your findings compare to those for traditional medication effects? I assume results are available for RCTs testing risperidone and aripiprazole.

Round 2

Reviewer 2 Report

Comments and Suggestions for Authors

The authors have addressed most of my concerns adequately. However, a limitation should be included in the discussion that information on potential confounding variables such as diet and medication use was not collected and these could have impacted microbiota profiles.

Comments on the Quality of English Language

Minor language errors need correction.

Author Response

Response to reviewer comment: Thank you for your useful comment. Please, check the revised manuscript in lines 347-352. Our manuscript was subjected to language correction by native speaker from English Editorial Services, Ltd., Brno, Czech Republic.